# CRITICAL LEARNING PERIODS IN DEEP NETWORKS

**Alessandro Achille** *
Department of Computer Science
University of California, Los Angeles
achille@cs.ucla.edu

**Matteo Rovere** *
Ann Romney Center for Neurologic Diseases
Brigham and Women's Hospital and Harvard Medical School
mrovere@bwh.harvard.edu

**Stefano Soatto**
Department of Computer Science
University of California, Los Angeles
soatto@cs.ucla.edu

## ABSTRACT

Similar to humans and animals, deep artificial neural networks exhibit critical periods during which a temporary stimulus deficit can impair the development of a skill. The extent of the impairment depends on the onset and length of the deficit window, as in animal models, and on the size of the neural network. Deficits that do not affect low-level statistics, such as vertical flipping of the images, have no lasting effect on performance and can be overcome with further training. To better understand this phenomenon, we use the Fisher Information of the weights to measure the effective connectivity between layers of a network during training. Counterintuitively, information rises rapidly in the early phases of training, and then decreases, preventing redistribution of information resources in a phenomenon we refer to as a loss of "Information Plasticity". Our analysis suggests that the first few epochs are critical for the creation of strong connections that are optimal relative to the input data distribution. Once such strong connections are created, they do not appear to change during additional training. These findings suggest that the initial learning transient, under-scrutinized compared to asymptotic behavior, plays a key role in determining the outcome of the training process. Our findings, combined with recent theoretical results in the literature, also suggest that forgetting (decrease of information in the weights) is critical to achieving invariance and disentanglement in representation learning. Finally, critical periods are not restricted to biological systems, but can emerge naturally in learning systems, whether biological or artificial, due to fundamental constrains arising from learning dynamics and information processing.

## 1 INTRODUCTION

Critical periods are time windows of early post-natal development during which sensory deficits can lead to permanent skill impairment (Kandel et al., 2013). Researchers have documented critical periods affecting a range of species and systems, from visual acuity in kittens (Wiesel & Hubel, 1963b; Wiesel, 1982) to song learning in birds (Konishi, 1985). Uncorrected eye defects (*e.g.*, strabismus, cataracts) during the critical period for visual development lead to amblyopia in one in fifty adults.

The cause of critical periods is ascribed to the biochemical modulation of windows of neuronal plasticity (Hensch, 2004). In this paper, however, we show that deep neural networks (DNNs), while completely devoid of such regulations, respond to sensory deficits in ways similar to those observed in humans and animal models. This surprising result suggests that critical periods may arise from information processing, rather than biochemical, phenomena.

We propose using the information in the weights, measured by an efficient approximation of the Fisher Information, to study critical period phenomena in DNNs. We show that, counterintuitively, the information in the weights does not increase monotonically during training. Instead, a rapid growth in information ("memorization phase") is followed by a *reduction of information* ("reorganization" or "forgetting" phase), even as classification performance keeps increasing. This behavior is consistent across different tasks and network architectures. Critical periods are centered in the memorization phase.

---

*These authors contributed equally to this work.

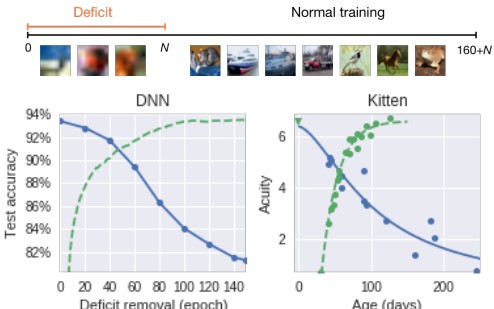 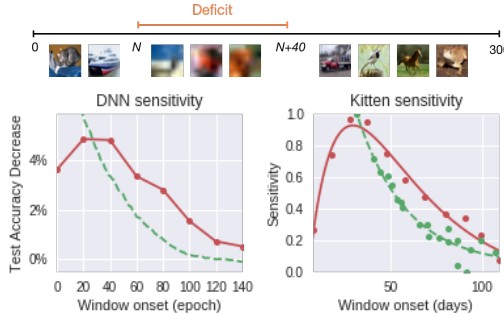

Figure 1: **DNNs exhibit critical periods.** **(A)** Final accuracy achieved by a CNN trained with a cataract-like deficit as a function of the training epoch $N$ at which the deficit is removed (solid line). Performance is permanently impaired if the deficit is not corrected early enough, regardless of how much additional training is performed. As in animal models, critical periods coincide with the early learning phase during which, in the absence of deficits, test accuracy would rapidly increase (dashed). **(B)** For comparison, we report acuity for kittens monocularly deprived since birth and tested at the time of eye-opening (solid), and normal visual acuity development (in kittens) as a function of their age (dashed) (Giffin & Mitchell, 1978; Mitchell, 1988). **Sensitivity during learning: (C)** Final test accuracy of a DNN as a function of the onset of a short 40-epoch deficit. The decrease in the final performance can be used to measure the sensitivity to deficits. The most sensitive epochs corresponds to the early rapid learning phase, before the test error (dashed line) begins to plateau. Afterwards, the network is largely unaffected by the temporary deficit. **(D)** This can be compared with changes in the degree of functional disconnection (normalized numbers of V1 monocular cells disconnected from the contralateral eye) as a function of the kittens' age at the onset of a 10-12-day deficit window (Olson & Freeman, 1980). Dashed lines are as in A and B respectively, up to a re-scaling of the y-axis.

Our findings, described in Section 2, indicate that the early transient is critical in determining the final solution of the optimization associated with training an artificial neural network. In particular, the effects of sensory deficits during a critical period cannot be overcome, no matter how much additional training is performed. Yet most theoretical studies have focused on the network behavior after convergence (Representation Learning) or on the asymptotic properties of the optimization scheme used for training (SGD).

To study this early phase, in Section 3, we use the Fisher Information to quantify the *effective connectivity* of a network during training, and introduce the notion of *Information Plasticity* in learning. Information Plasticity is maximal during the memorization phase, and decreases in the reorganization phase. We show that deficit sensitivity during critical periods correlates strongly with the effective connectivity.

In Section 4 we discuss our contribution in relation to previous work. When considered in conjunction with recent results on representation learning (Achille & Soatto, 2018), our findings indicate that forgetting (reducing information in the weights) is critical to achieving invariance to nuisance variability as well as independence of the components of the representation, but comes at the price of reduced adaptability later in the training. We also hypothesize that the loss of physical connectivity in biology (neural plasticity) could be a consequence, rather than a cause, of the loss of Information Plasticity, which depends on how the information is distributed throughout a network during the early stages of learning. These results also shed light on the common practice of pre-training a model on a task and then fine-tune it for another, one of the most rudimentary forms of transfer learning. Our experiments show that, rather than helpful, pre-training can be detrimental, even if the tasks are similar (*e.g.*, same labels, slightly blurred images).

## 2 EXPERIMENTS

A notable example of critical period-related deficit, commonly affecting humans, is amblyopia (reduced visual acuity in one eye) caused by cataracts during infancy or childhood (Taylor et al., 1979;

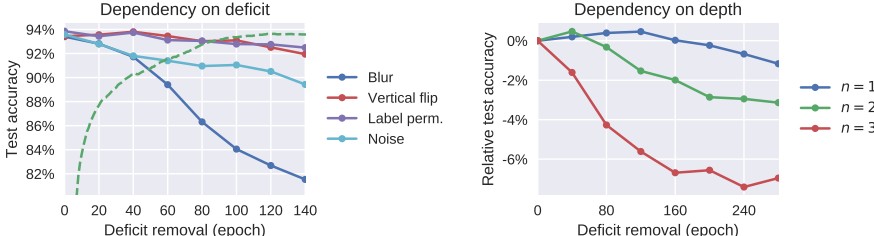

Figure 2: **(Left) High-level perturbations do not induce a critical period.** When the deficit only affects high-level features (vertical flip of the image) or the last layer of the CNN (label permutation), the network does not exhibit critical periods (test accuracy remains largely flat). On the other hand, a sensory deprivation-like deficit (image is replaced by random noise) does cause a deficit, but the effect is less severe than in the case of image blur. **(Right) Dependence of the critical period profile on the network's depth.** Adding more convolutional layers increases the effect of the deficit during its critical period (shown here is the decrease in test accuracy due to the deficit with respect to the test accuracy reached without deficits).

von Noorden, 1981). Even after surgical correction of cataracts, the ability of the patients to regain normal acuity in the affected eye depends both on the duration of the deficit and on its age of onset, with earlier and longer deficits causing more severe effects. In this section, we aim to study the effects of similar deficits in DNNs. To do so, we train a standard All-CNN architecture based on Springenberg et al. (2014) (see Appendix A) to classify objects in small $32 \times 32$ images from the CIFAR-10 dataset (Krizhevsky & Hinton, 2009). We train with SGD using an exponential annealing schedule for the learning rate. To simulate the effect of cataracts, for the first $t_0$ epochs the images in the dataset are downsampled to $8 \times 8$ and then upsampled back to $32 \times 32$ using bilinear interpolation, in practice blurring the image and destroying small-scale details.[1] After that, the training continues for 160 more epochs, giving the network time to converge and ensuring it is exposed to the same number of uncorrupted images as in the control ($t_0 = 0$) experiment.

**DNNs exhibit critical periods:** In Figure 1, we plot the final performance of a network affected by the deficit as a function of the epoch $t_0$ at which the deficit is corrected. We can readily observe the existence of a critical period: If the blur is not removed within the first 40-60 epochs, the final performance is severely decreased when compared to the baseline (up to a threefold increase in error). The decrease in performance follows trends commonly observed in animals, and may be qualitatively compared, for example, to the loss of visual acuity observed in kittens monocularly deprived from birth as a function of the length of the deficit (Mitchell, 1988).[2]

We can measure more accurately the sensitivity to a blur deficit during learning by introducing the deficit in a short window of constant length (40 epochs), starting at different epochs, and then measure the decrease in the DNN's final performance compared to the baseline (Figure 1). Doing this, we observe that the sensitivity to the deficit peaks in the central part of the early rapid learning phase (at around 30 epochs), while introducing the deficit later produces little or no effect. A similar experiment performed on kittens, using a window of 10-12 days during which the animals are monocularly deprived, again shows a remarkable similarity between the profiles of the sensitivity curves (Olson & Freeman, 1980).

**High-level deficits are not associated with a critical period:** A natural question is whether any change in the input data distribution will have a corresponding critical period for learning. This is not the case for neuronal networks, which remain plastic enough to adapt to high-level changes in sensory processing (Daw, 2014). For example, it is well-reported that even adult humans can rapidly adapt to certain drastic changes, such as the inversion of the visual field (Stratton, 1896; Kohler, 1964). In Figure 2, we observe that DNNs are also largely unaffected by high-level deficits – such as vertical flipping of the image, or random permutation of the output labels: After deficit correction, the network quickly recovers its baseline performance. This hints at a finer interplay between the

---

[1]We employed this method, instead of a simpler Gaussian blur, since it has a very similar effect and makes the quantification of information loss clearer.

[2]See Appendix C for details on how to compare different models and deficits.

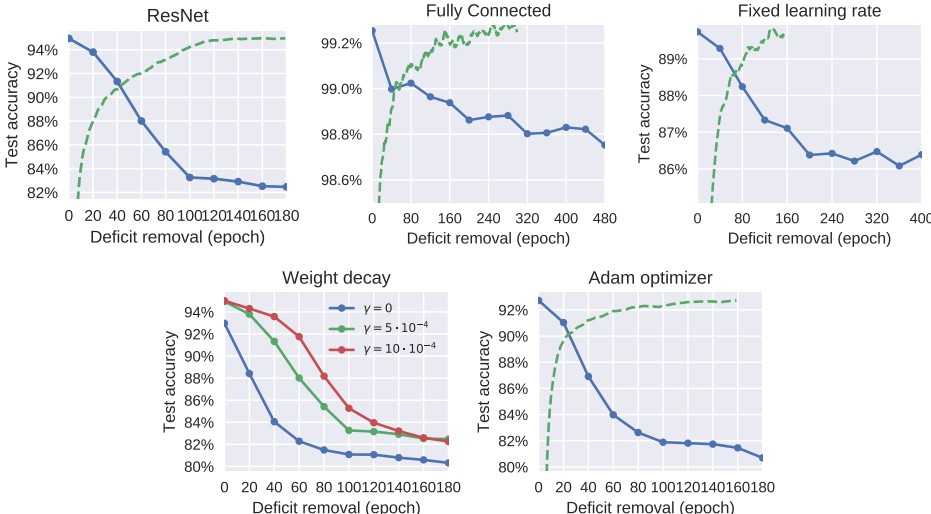

Figure 3: **Critical periods in different DNN architectures and optimization schemes. (Left)** Effect of an image blur deficit in a ResNet architecture trained on CIFAR-10 with learning rate annealing and **(Center)** in a deep fully-connected network trained on MNIST with a fixed learning rate. Different architectures, using different optimization methods and trained on different datasets, still exhibit qualitatively similar critical period behavior. **(Right)** Same experiment as in Figure 1, but using a fixed learning rate instead of an annealing scheme. Although the time scale of the critical period is longer, the trends are similar, supporting the notion that critical periods cannot be explained solely in terms of the loss landscape of the optimization. **(Bottom Left)** Networks trained without weight decay have shorter and sharper critical periods. Gradually increasing the weight decay makes the critical period longer, until the point where it stops training properly. **(Bottom Right)** Using a different optimization method (Adam) we observe a similar behavior to standard SGD.

structure of the data distribution and the optimization algorithm, resulting in the existence of a critical period.

**Sensory deprivation:** We now apply to the network a more drastic deficit, where each image is replaced by white noise. Figure 2 shows hows this extreme deficit exhibits a remarkably less severe effect than the one obtained by only blurring images: Training the network with white noise does not provide any information on the natural images, and results in milder effects than those caused by a deficit (*e.g.*, image blur), which instead conveys *some* information, but leads the network to (incorrectly) learn that no fine structure is present in the images. A similar effect has been observed in animals, where a period of early sensory deprivation (dark-rearing) can lengthen the critical period and thus cause less severe effects than those documented in light-reared animals (Mower, 1991). We refer the reader to Appendix C for a more detailed comparison between sensory deprivation and training on white noise.

**Architecture, depth, and learning rate annealing:** Figure 3 shows that a fully-connected network trained on the MNIST digit classification dataset also shows a critical period for the image blur deficit. Therefore, the convolutional structure is not necessary, nor is the use of natural images. Similarly, a ResNet-18 trained on CIFAR-10 also has a critical period, which is also remarkably sharper than the one found in a standard convolutional network (Figure 1). This is especially interesting, since ResNets allow for easier backpropagation of gradients to the lower layers, thus suggesting that the critical period is not caused by vanishing gradients. However, Figure 2 (Right) shows that the presence of a critical period does indeed depend critically on the depth of the network. In Figure 3, we confirm that a critical period exists even when the network is trained with a constant learning rate, and therefore cannot be explained by an annealed learning rate in later epochs.

**Optimization method and weight decay:** Figure 3 (Bottom Right) shows that when using Adam as the optimization scheme, which renormalizes the gradients using a running mean of their first two moments, we still observe a critical period similar to that of standard SGD. However, changing the

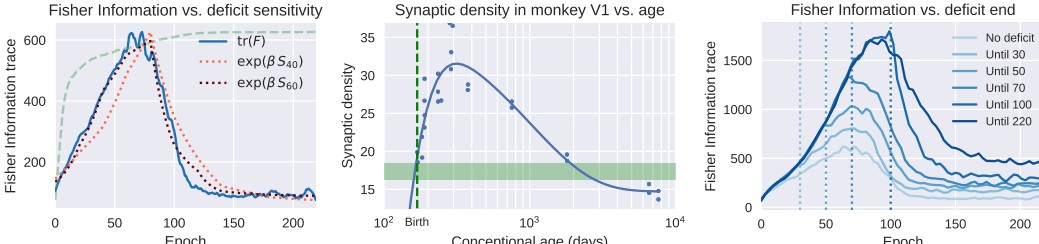

Figure 4: **Critical periods in DNNs are traced back to changes in the Fisher Information.** **(Left)** Trace of the Fisher Information of the network weights as a function of the training epoch (blue line), showing two distinct phases of training: First, information sharply increases, but once test performance starts to plateau (green line), the information in the weights decreases during a "consolidation" phase. Eventually less information is stored, yet test accuracy improves slightly (green line). The weights' Fisher Information correlates strongly with the networks sensitivity to critical periods, computed as in Figure 1 using both a window size of 40 and 60, and fitted here to the Fisher Information using a simple exponential fit. **(Center)** Recalling the connection between FIM ad connectivity, we may compare it to synaptic density during development in the visual cortex of macaques (Rakic et al., 1986). Here too, a rapid increase in connectivity is followed by elimination of synapses (pruning) continuing throughout life. **(Right)** Effects of critical period-inducing blurring on the Fisher Information: The impaired network uses more information to solve the task, compared to training in the absence of a deficit, since it is forced to memorize the labels case by case.

hyperparameters of the optimization can change the shape of the critical period: In Figure 3 (Bottom Left) we show that increasing weight decay makes critical periods longer and less sharp. This can be explained as it both slows the convergence of the network, and it limits the ability of higher layers to change to overcome the deficit, thus encouraging lower layers to also learn new features.

## 3 FISHER INFORMATION ANALYSIS

We have established empirically that, in animals and DNNs alike, the initial phases of training are critical to the outcome of the training process. In animals, this strongly relates to changes in the brain architecture of the areas associated with the deficit (Daw, 2014). This is inevitably different in artificial networks, since their connectivity is formally fixed at all times during training. However, not all the connections are equally useful to the network: Consider a network encoding the approximate posterior distribution $p_w(y|x)$, parameterized by the weights $w$, of the task variable $y$ given an input image $x$. The dependency of the final output from a specific connection can be estimated by perturbing the corresponding weight and looking at the magnitude of the change in the final distribution. Specifically, given a perturbation $w' = w + \delta w$ of the weights, the discrepancy between the $p_w(y|x)$ and the perturbed network output $p_{w'}(y|x)$ can be measured by their Kullback-Leibler divergence, which, to second-order approximation, is given by:

$$\mathbb{E}_x \operatorname{KL}(\, p_{w'}(y|x) \,\|\, p_w(y|x) \,) = \delta w \cdot F \delta w + o(\delta w^2),$$

where the expectation over $x$ is computed using the empirical data distribution $\hat{Q}(x)$ given by the dataset, and

$$F := \mathbb{E}_{x \sim \hat{Q}(x)} \mathbb{E}_{y \sim p_w(y|x)} [\nabla_w \log p_w(y|x) \nabla_w \log p_w(y|x)^T]$$

is the Fisher Information Matrix (FIM). The FIM can thus be considered a local metric measuring how much the perturbation of a single weight (or a combination of weights) affects the output of the network (Amari & Nagaoka, 2000). In particular, weights with low Fisher Information can be changed or "pruned" with little effect on the network's performance. This suggests that the Fisher Information can be used as a measure of the effective connectivity of a DNN, or, more generally, of the "synaptic strength" of a connection (Kirkpatrick et al., 2017). Finally, the FIM is also a semi-definite approximation of the Hessian of the loss function (Martens, 2014) and hence of the curvature of the loss landscape at a particular point $w$ during training, providing an elegant connection between the FIM and the optimization procedure (Amari & Nagaoka, 2000), which we will also employ later.

Unfortunately, the full FIM is too large to compute. Rather, we use its trace to measure the global or layer-wise connection strength, which we can compute efficiently using (Appendix A):

$$\mathrm{tr}(F) = \mathbb{E}_{x \sim \hat{Q}(x)} \mathbb{E}_{y \sim p_w(y|x)} [\|\nabla_w \log p_w(y|x)\|^2].$$

In order to capture the behavior of the off-diagonal terms, we also tried computing the log-determinant of the full matrix using the Kronecker-Factorized approximation of Martens & Grosse (2015), but we observed the same qualitative trend as the trace. Since the FIM is a local measure, it is very sensitive to the irregularities of the loss landscape. Therefore, in this section we mainly use ResNets, which have a relatively smooth landscape (Li et al., 2018). For other architectures we use instead a more robust estimator of the FIM based on the injection of noise in the weights (Achille & Soatto, 2018), also described in Appendix A.

**Two phases of learning:** As its name suggests, the FIM can be thought as a measure of the quantity of information about the training data that is contained in the model (Fisher, 1925). Based on this, one would expect the overall strength of the connections to increase monotonically as we acquire information from experience. However, this is not the case: While during an initial phase the network acquires information about the data, which results in a large increase in the strength of the connections, once the performance in the task begins to plateau, the network starts decreasing the overall strength of its connections. However, this does not correspond to a reduction in performance, rather, performance keeps slowly improving. This can be seen as a "forgetting, or "compression" phase, during which redundant connections are eliminated and non-relevant variability in the data is discarded. It is well-established how the elimination ("pruning") of unnecessary synapses is a fundamental process during learning and brain development (Rakic et al., 1986) (Figure 4, Center); in Figure 4 (Left) an analogous phenomenon is clearly and quantitatively shown for DNNs.

Strikingly, these changes in the connection strength are closely related to the sensitivity to critical-period-inducing deficits such as image blur, computed using the "sliding window" method as in Figure 1. In Figure 4 we see that the sensitivity closely follows the trend of the FIM. This is remarkable since the FIM is a local quantity computed at a single point during the training of a network in the absence of deficit, while sensitivity during a critical period is computed, using test data, at the end of the impaired network training. Figure 4 (Right) further emphasizes the effect of deficits on the FIM: in the presence of a deficit, the FIM grows and remains substantially higher even after the deficit is removed. This may be attributed to the fact that, when the data are so corrupted that classification is impossible, the network is forced to memorize the labels, therefore increasing the quantity of information needed to perform the same task.

**Layer-wise effects of deficits:** A layer-wise analysis of the FIM sheds further light on how the deficit affects the network. When the network (in this case All-CNN, which has a clearer division among layers than ResNet) is trained without deficits, the most important connections are in the intermediate layers (Figure 5, Left), which can process the input CIFAR-10 image at the most informative intermediate scale. However, if the network is initially trained on blurred data (Figure 5, top right), the strength of the connections is dominated by the top layer (Layer 6). This is to be expected, since the low-level and mid-level structures of the images are destroyed, making the lower layers ineffective. However, if the deficit is removed early in the training (Figure 5, top center), the network manages to "reorganize", reducing the information contained in the last layer, and, at the same time, increasing the information in the intermediate layers. We refer to these phenomena as changes in "Information Plasticity". If, however, the data change occurs after the consolidation phase, the network is unable to change its effective connectivity: The connection strength of each layer remains substantially constant. The network has lost its Information Plasticity and is past its critical period.

**Critical periods as bottleneck crossings:** The analysis of the FIM also sheds light on the geometry of the loss function and the learning dynamics. Since the FIM can be interpreted as the local curvature of the residual landscape, Fig. 4 shows that learning entails crossing bottlenecks: In the initial phase the network enters regions of high curvature (high Fisher Information), and once consolidation begins, the curvature decreases, allowing it to cross the bottleneck and enter the valley below. If the statistics change after crossing the bottleneck, the network is trapped. In this interpretation, the early phases of convergence are critical in leading the network towards the "right" final valley. The end of critical periods comes after the network has crossed all bottlenecks (and thus learned the features) and entered a wide valley (region of the weight space with low curvature, or low Fisher Information).

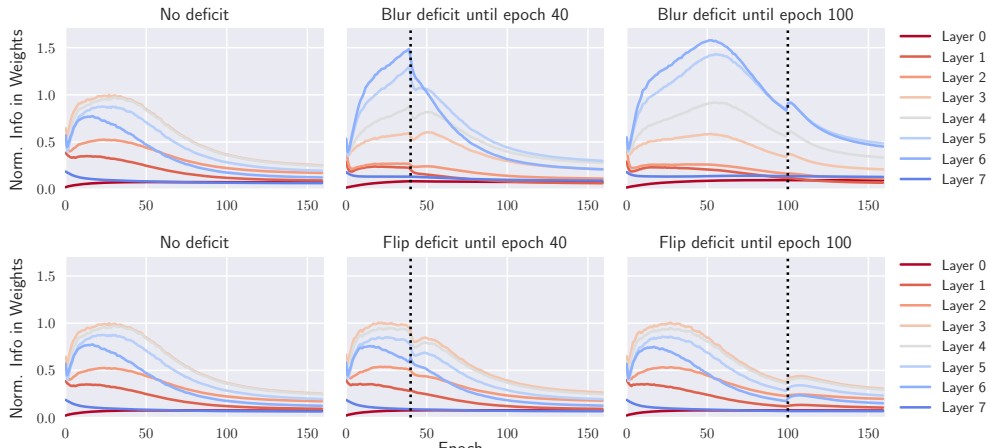

Figure 5: Normalized quantity of information contained in the weights of each layer as a function of the training epoch. **(Top Left)** In the absence of deficits, the network relies mostly on the middle layers (3-4-5) to solve the task. **(Top Right)** In the presence of an image blur deficit until epoch 100, more resources are allocated to the higher layers (6-7) rather than to the middle layers. The blur deficit destroys low- and mid-level features processed by those layers, leaving only the global features of the image, which are processed by the higher layers. Even if the deficit is removed, the middle layers remain underdeveloped. **(Top Center)** When the deficit is removed at an earlier epoch, the layers can partially reconfigure (notice, *e.g.*, the fast loss of information of layer 6), resulting in less severe long-term consequences. We refer to the redistribution of information and the relative changes in effective connectivity as "Information Plasticity". **(Bottom row)** Same plots, but using a vertical flip deficit, which does not induce a critical period. As expected, the quantity of information in the layers is not affected.

## 4 DISCUSSION AND RELATED WORK

Critical periods have thus far been considered an exclusively biological phenomenon. At the same time, the analysis of DNNs has focused on asymptotic properties and neglected the initial transient behavior. To the best of our knowledge, we are the first to show that artificial neural networks exhibit critical period phenomena, and to highlight the critical role of the transient in determining the asymptotic performance of the network. Inspired by the role of synaptic connectivity in modulating critical periods, we introduce the use of Fisher Information to study this initial phase. We show that the initial sensitivity to deficits closely follows changes in the FIM, both global, as the network first rapidly increases and then decreases the amount of stored information, and layer-wise, as the network "reorganizes" its effective connectivity in order to optimally process information.

Our work naturally relates to the extensive literature on critical periods in biology. Despite artificial networks being an extremely reductionist approximation of neuronal networks, they exhibit behaviors that are qualitatively similar to the critical periods observed in human and animal models. Our information analysis shows that the initial rapid memorization phase is followed by a loss of Information Plasticity which, counterintuitively, further improves the performance. On the other hand, when combined with the analysis of Achille & Soatto (2018) this suggests that a "forgetting" phase may be desirable, or even necessary, in order to learn robust, nuisance-invariant representations.

The existence of two distinct phases of training has been observed and discussed by Shwartz-Ziv & Tishby (2017), although their analysis builds on the (Shannon) information of the *activations*, rather than the (Fisher) information in the *weights*. On a multi-layer perceptron (MLP), Shwartz-Ziv & Tishby (2017) empirically link the two phases to a sudden increase in the gradients' covariance. It may be tempting to compare these results with our Fisher Information analysis. However, it must be noted that the FIM is computed using the gradients with respect to the model prediction, not to the ground truth label, leading to important qualitative differences. In Figure 6, we show that the covariance and norm of the gradients exhibit no clear trends during training with and without deficits, and, therefore, unlike the FIM, do not correlate with the sensitivity to critical periods. However,

a connection between our FIM analysis and the information in the activations can be established based on the work of Achille & Soatto (2018), which shows that the FIM of the weights can be used to bound the information in the activations. In fact, we may intuitively expect that pruning of connections naturally leads to loss of information in the corresponding activations. Thus, our analysis corroborates and expands on some of the claims of Shwartz-Ziv & Tishby (2017), while using an independent framework.

Aside from being more closely related to the deficit sensitivity during critical periods, Fisher's Information also has a number of technical advantages: Its diagonal is simple to estimate, even on modern state-of-the-art architectures and compelling datasets, and it is less sensitive to the choice estimator of mutual information, avoiding some of the common criticisms to the use of information quantities in the analysis of deep learning models. Finally, the FIM allows us to probe fine changes in the effective connectivity across the layers of the network (Figure 5), which are not visible in Shwartz-Ziv & Tishby (2017).

A complete analysis of the activations should account not only for the amount of information (both task- and nuisance-related), but also for its accessibility, *e.g.*, how easily task-related information can be extracted by a linear classifier. Following a similar idea, Montavon et al. (2011) aim to study the layer-wise, or "spatial" (but not temporal) evolution of the simplicity of the representation by performing a principal component analysis (PCA) of a radial basis function (RBF) kernel embedding of each layer representation. They show that, on a multi-layer perceptron, task-relevant information increasingly concentrate on the first principal components of the representation's embedding, implying that they become more easily "accessible" layer after layer, while nuisance information (when it is codified at all) is encoded in the remaining components. In our work we instead focus on the temporal evolution of the weights. However, it's important to notice that a network with simpler weights (as measured by the FIM) also requires a simpler smooth representation (as measured, *e.g.*, by the RBF embedding) in order to operate properly, since it needs to be resistant to perturbations of the weights. Thus our analysis is wholly compatible with the intuitions of Montavon et al. (2011). It would also be interesting to study the joint spatio-temporal evolution of the network using both frameworks at once.

One advantage of focusing on the information of the weights rather than on the activations, or behavior of the network, is to have a readout of the "effective connectivity" during critical periods, which can be compared to similar readouts in animals. In fact, "behavioral" readouts upon deficit removal, both in artificial and neuronal networks, can potentially be confounded by deficit-coping changes at different levels of the visual pathways (Daw, 2014; Knudsen, 2004). On the other hand, deficits in deprived animals are mirrored by abnormalities in the circuitry of the visual pathways, which we characterize in DNNs using the FIM to study its "effective connectivity", *i.e.*, the connections that are actually employed by the network to solve the task. Sensitivity to critical periods and the trace of the Fisher Information peak at the same epochs, in accord with the evidence that skill development and critical periods in neuronal networks are modulated by changes (generally experience-dependent) in synaptic plasticity (Knudsen, 2004; Hensch, 2004). Our layer-wise analysis of the Fisher Information (Figure 5) also shows that visual deficits reinforce higher layers to the detriment of intermediate layers, leaving low-level layers virtually untouched. If the deficit is removed after the critical period ends, the network is not able to reverse these effects. Although the two systems are radically different, a similar response can be found in the visual pathways of animal models: Lower levels (*e.g.*, retina, lateral geniculate nucleus) and higher-level visual areas (*e.g.*, V2 and post-V2) show little remodeling upon deprivation, while most changes happen in different layers of V1 (Wiesel & Hubel, 1963a; Hendrickson et al., 1987).

An insightful interpretation of critical periods in animal models was proposed by Knudsen (2004): The initial connections of neuronal networks are unstable and easily modified (highly plastic), but as more "samples" are observed, they change and reach a more stable configuration which is difficult to modify. Learning can, however, still happen within the newly created connectivity pattern. This is largely compatible with our findings: Sensitivity to critical-period-inducing deficits peaks when connections are remodeled (Figure 4, Left), and different connectivity profiles are observed in networks trained with and without a deficit (Figure 5). Moreover, high-level deficits such as image-flipping and label permutation, which do not require restructuring of the network's connections in order to be corrected, do not exhibit a critical period.

Applying a deficit at the beginning of the training may be compared to the common practice of pre-training, which is generally found to improve the performance of the network. Erhan et al. (2010) study the somewhat related, but now seldom used, practice of layer-wise unsupervised pre-training, and suggest that it may act as a regularizer by moving the weights of the network towards an area of the loss landscape closer to the attractors for good solutions, and that early examples have a stronger effect in steering the network towards particular solutions. Here, we have shown that pre-training on blurred data can have the opposite effect; *i.e.*, it can severely decrease the final performance of the network. However, in our case, interpreting the deficits effect as moving the network close to a bad attractor is difficult to reconcile with the smooth transition observed in the critical periods, since the network would either converge to this attractor, and thus have low accuracy, or escape completely.

Instead, we reconcile our experiments with the geometry of the loss function by introducing a different explanation based on the interpretation of the FIM as an approximation of the local curvature. Figure 4 suggests that SGD encounters two different phases during the network training: At first, the network moves towards high-curvature regions of the loss landscape, while in the second phase the curvature decreases and the network eventually converges to a flat minimum (as observed in Keskar et al. (2017)). We can interpret these as the network crossing narrow bottlenecks during its training in order to learn useful features, before eventually entering a flat region of the loss surface once learning is completed and ending up trapped there. When combining this assumption with our deficit sensitivity analysis, we can hypothesize that the critical period occurs precisely upon crossing of this bottleneck. It is also worth noticing how there is evidence that convergence to flat minima (minima with low curvature) in a DNN correlates with a good generalization performance (Hochreiter & Schmidhuber, 1997; Li et al., 2018; Chaudhari et al., 2017; Keskar et al., 2017). Indeed, using this interpretation, Figure 4 (Right) tells us that networks more affected by the deficit converge to sharper minima. However, we have also found that the performance of the network is already mostly determined during the early "sensitive" phase. The final sharpness at convergence may therefore be an epiphenomenon, rather than the cause of good generalization.

## 5 CONCLUSION

Our goal in this paper is not so much to investigate the human (or animal) brain through artificial networks, as to understand fundamental information processing phenomena, both in their biological or artificial implementations. It is also not our goal to suggest that, since they both exhibit critical periods, DNNs are necessarily a valid model of neurobiological information processing, although recent work has emphasized this aspect. We engage in an "Artificial Neuroscience" exercise in part to address a technological need to develop "explainable" artificial intelligence systems whose behavior can be understood and predicted. While traditionally well-understood mathematical models were used by neuroscientists to study biological phenomena, information processing in modern artificial networks is often just as poorly understood as in biology, so we chose to exploit well-known biological phenomena as probes to study information processing in artificial networks.

Conversely, it would also be interesting to explore ways to test whether biological networks prune connections as a consequences of a loss of Information Plasticity, rather than as a cause. The mechanisms underlying network reconfiguration during learning and development might be an evolutionary outcome obtained under the pressure of fundamental information processing phenomena.

### ACKNOWLEDGEMENTS

We thank the anonymous reviewers for their thoughtful feedback, and for suggesting new experiments and relevant literature. Supported by ONR N00014-17-1-2072, ARO W911NF-17-1-0304, AFOSR FA9550-15-1-0229 and FA8650-11-1-7156.

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

# A  DETAILS OF THE EXPERIMENTS

## A.1  ARCHITECTURES AND TRAINING

In all of the experiments, unless otherwise stated, we use the following All-CNN architecture, adapted from Springenberg et al. (2014):

```
conv 96 - conv 96 - conv 192 s2 - conv 192 - conv 192 - conv 192
  s2 - conv 192 - conv1 192 - conv1 10 - avg. pooling - softmax
```

where each `conv` block consists of a $3 \times 3$ convolution, batch normalization and ReLU activations. `conv1` denotes a $1 \times 1$ convolution. The network is trained with SGD, with a batch size of 128, learning rate starting from 0.05 and decaying smoothly by a factor of .97 at each epoch. We also use weight decay with coefficient 0.001. In the experiments with a fixed learning rate, we fix the learning rate to 0.001, which we find to allow convergence without excessive overfitting. For the ResNet experiments, we use the ResNet-18 architecture from He et al. (2016) with initial learning rate 0.1, learning rate decay .97 per epoch, and weight decay 0.0005. When training with Adam, we use a learning rate of 0.001 and weight decay 0.0001.

When experimenting with varying network depths, we use the following architecture:

```
conv 96 - [conv 96·2^{i-1} - conv 96·2^i s2]_{i=1}^n - conv 96·2^n - conv1 96·2^n
                            - conv1 10
```

In order to avoid interferences between the annealing scheme and the architecture, in these experiments we fix the learning rate to 0.001.

The Fully Connected network used for the MNIST experiments has hidden layers of size $[2500, 2000, 1500, 1000, 500]$. All hidden layers use batch normalization followed by ReLU activations. We fix the learning rate to 0.005. Weight decay is not used. We use data augmentation with random translations up to 4 pixels and random horizontal flipping. For MNIST, we pad the images with zeros to bring them to size $32 \times 32$.

## A.2  APPROXIMATIONS OF THE FISHER INFORMATION MATRIX

To compute the trace of the Fisher Information Matrix, we use the following expression derived directly from the definition:

$$\text{tr}(F) = \mathbb{E}_{x \sim \hat{Q}(x)} \mathbb{E}_{y \sim p_w(y|x)} [\text{tr}(\nabla_w \log p_w(y|x) \nabla_w \log p_w(y|x)^T)]$$
$$= \mathbb{E}_{x \sim \hat{Q}(x)} \mathbb{E}_{y \sim p_w(y|x)} [\|\nabla_w \log p_w(y|x)\|^2],$$

where the input image $x$ is sampled from the dataset, while the label $y$ is sampled from the output posterior. Expectations are approximated by Monte-Carlo sampling. Notice, however, that this expression depends only on the local gradients of the loss with respect to the weights at a point $w = w_0$, so it can be noisy when the loss landscape is highly irregular. This is not a problem for ResNets Li et al. (2018), but for other architectures we use instead a different technique, proposed in Achille & Soatto (2018). More in detail, let $L(w)$ be the standard cross-entropy loss. Given the current weights $w_0$ of the network, we find the diagonal matrix $\Sigma$ that minimizes:

$$L' = \mathbb{E}_{w \sim N(w_0, \Sigma)}[L(w)] - \beta \log |\Sigma|,$$

where $\beta$ is a parameter that controls the smoothness of the approximation. Notice that $L'$ can be minimized efficiently using the method in Kingma et al. (2015). To see how this relates to the Fisher Information Matrix, assume that $L(w)$ can be approximated locally in $w_0$ as $L(w) = L_0 + a \cdot w + w \cdot Hw$. We can then rewrite $L'$ as

$$L' = L_0 + \text{tr}(\Sigma H) - \beta \log |\Sigma|.$$

Taking the derivative with respect to $\Sigma$, and setting it to zero, we obtain $\Sigma_{ii} = \beta/H_{ii}$. We can then use $\Sigma$ to estimate the trace of the Hessian, and hence of the Fisher information.

## A.3 CURVE FITTING

Fitting of sensitivity curves and synaptic density profiles from the literature was performed using:

$$f(t) = e^{-(t-d)/\tau_1} - ke^{-(t-d)/\tau_2}$$

as the fitting equation, where $t$ is the age at the time of sampling and $\tau_1$, $\tau_2$, $k$ and $d$ are unconstrained parameters (Banks et al., 1975).

The exponential fit of the sensitivity to the Fisher Information trace uses the expression

$$F(t) = a\exp(cS_k(t)) + b,$$

where $a$, $b$ and $c$ are unconstrained parameters, $F(t)$ is the Fisher Information trace at epoch $t$ of the training of a network without deficits and $S_k$ is the sensitivity computed using a window of size $k$. That is, $S_k(t)$ is the increase in the final test error over a baseline when the network is trained in the presence of a deficit between epochs $t$ and $t + k$.

## B   ADDITIONAL PLOTS

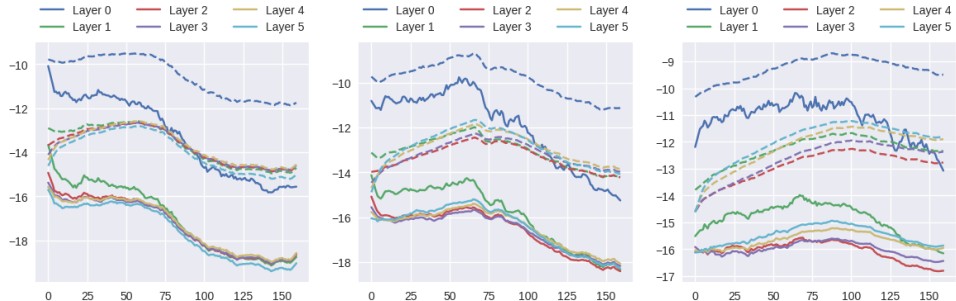

Figure 6: Log of the norm of the gradient means (solid line) and standard deviation (dashed line) during training when: **(Left)** No deficit is present, **(Center)** A blur deficit is present until epoch 70, and **(Right)** a deficit is present until the last epoch. Notice that the presence of a deficit does not decrease the magnitude of the gradients propagated to the first layers during the last epochs, rather it seems to increase it, suggesting that vanishing gradients are not the cause of the critical period for the blurring deficit.

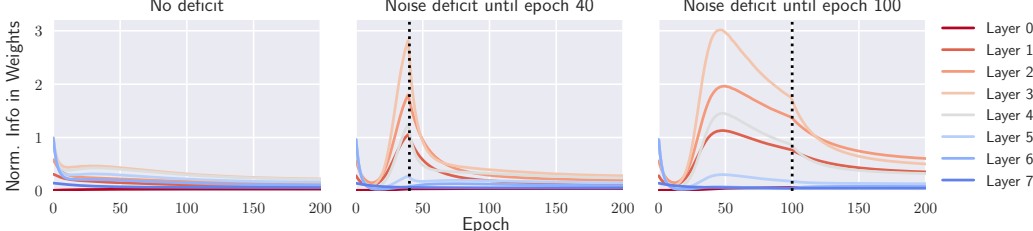

Figure 7: Same plot as in Figure 5, but for a noise deficit. Unlike with blur, much more resources are allocated to the lower-layers rather than higher-layers. This may explain why it is easier for the network to reconfigure to solve the task after the deficit is removed.

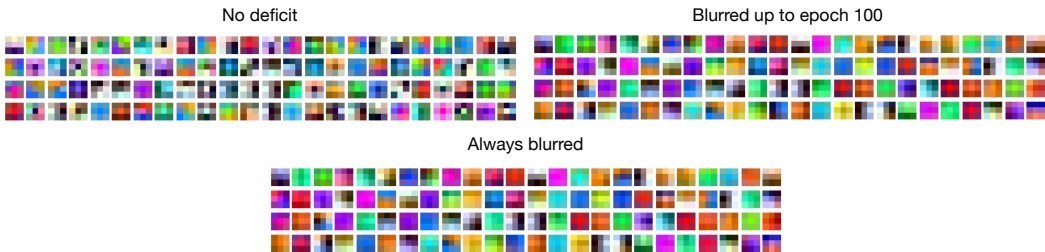

Figure 8: Visualization of the filters of the first layer of the network used for the experiment in Figure 1. In absence of a deficit, the network learns high-frequency filters, as seen by the fact that many filters are not smooth (first picture). However, when a blurring deficit is present, the network learns only smooth filters corresponding to low-frequencies of the input (third picture). If the deficit is removed after the end of the critical period, the network does not manage to learn high-frequency filters (second picture).

## C EXPERIMENTAL DESIGN AND COMPARISON WITH ANIMAL MODELS

Critical periods are task- and deficit-specific. The specific task we address is visual acuity, but the performance is necessarily measured through different mechanisms in animals and Artificial Neural Networks. In animals, visual acuity is traditionally measured by testing the ability to discriminate between black-and-white contrast gratings (with varying spatial frequency) and a uniform gray field. The outcome of such tests generally correlates well with the ability of the animal to use the eye to solve other visual tasks relying on acuity. Convolutional Neural Networks, on the other hand, have a very different sensory processing mechanism (based on heavily quantized data), which may trivialize such a test. Rather, we directly measure the performance of the network on an high-level task, specifically image classification, for which CNNs are optimized.

We chose to simulate cataracts in our DNN experiments, a deficit which allows us to explore its complex interactions with the structure of the data and the architecture of the network. Unfortunately, while the overall trends of cataract-induced critical periods have been studied and understood in animal models, there is not enough data to confidently regress sensibility curves comparable to those obtained in DNNs. For this reason, in Figure 1 we compare the performance loss in a DNN trained in the presence of a cataract-like deficit with the results obtained from monocularly deprived kittens, which exhibit similar trends and are one of the most common experimental paradigms in the visual neurosciences.

Simulating complete visual deprivation in a neural network is not as simple as feeding a constant stimulus: a network presented with a constant blank input will rapidly become trivial and thus unable to train on new data. This is to be expected, since a blank input is a perfectly predictable stimulus and thus the network can quickly learn the (trivial) solution to the task. We instead wanted to model an uninformative stimulus, akin to noise. Moreover, even when the eyes are sutured or maintained in the darkness, there will be background excitation of photoreceptors that is best modeled as noise. To account for this, we simulate sensory deprivation by replacing the input images with a dataset composed of (uninformative) random Gaussian noise. This way the network is trained on solving the highly non-trivial task of memorizing the association between the finitely-many noise patterns and their corresponding labels.

