# OpenReview forum: "Critical Learning Periods in Deep Networks"
_ICLR.cc/2019/Conference_

### Official Review · AnonReviewer1 · 2018-10-20
**Learning Phases in DNNs**

**Rating:** 6
**Confidence:** 5

**Review:**

The paper is interesting and I like it. I draws parallels from biological learning and the well known critical learning phases in biological systems to artificial neural network learning.
A series of empirical simulation experiments that all aim to disturb the learning process of the DNN and to artificially create criticality are presented. They are providing food for thought, in order to introduce some quantitative results, the authors use well known Fisher Information to measure the changes. So far so good and interesting.
I was disappointed to see Tishby's result (2017) only remotely discussed, an earlier work than the one by Tishby is by Montavon et al 2011 in JMLR. Also in this work properties of successive compression and dimensionality reduction are discussed, perhaps the starting point of quantitative analysis of various DNNs.

To this point the paper presents no theoretical contribution, rather empirical findings only, that may or may not be ubiquitous in DNN learning systems. The latter point may be worthwhile to discuss and analyse.
Overall, the paper is interesting with its nice empirical studies but stays somewhat superficial. To learn more a simpler toy model may be worthwhile to study.

---

> ### Author Response · Authors · 2018-11-21
> **Response to Reviewer 1**
>
> We thank the reviewer for their feedback and suggestions. We have updated the paper accordingly, and address some of the points more in detail below:
>
> >> I was disappointed to see Tishby's result (2017) only remotely discussed, an earlier work than the one by Tishby is by Montavon et al 2011 in JMLR. Also in this work properties of successive compression and dimensionality reduction are discussed, perhaps the starting point of quantitative analysis of various DNNs.
>
> We preferred not to elaborate at length on the connections with Schwartz and Tishby's results, since the relationship between the FIM of the weights (which we use in our paper) and the Shannon information of the activations, used by Tishby, is non-trivial and has already been discussed in more detail by other authors. However, given how important this aspect is and since it has also been discussed by the other reviewers, we have included in the revised version an extended discussion on it, which hopefully will also make the manuscript more self-contained.
>
> Concerning Montavon's paper, that is indeed our miss; we have added the paper to the revised discussion, and thank the reviewer for pointing it out.
>
> >> To this point the paper presents no theoretical contribution, rather empirical findings only, that may or may not be ubiquitous in DNN learning systems. The latter point may be worthwhile to discuss and analyse.
> Overall, the paper is interesting with its nice empirical studies but stays somewhat superficial.
>
> The empirical findings are observed across the most commonly used architectures and optimization algorithms, as we also confirm with the new experiments in Fig. 3. But it is true that it will take much more experimentation to assess whether they are truly ubiquitous and how they may affect different kinds of data. On the theoretical side, the analysis of the transient, irreversible, properties of the learning process using the Fisher information in the weights is not only novel, but also different from other theoretical analyses, such as the study of flat minima, which focuses on the asymptotic behavior of the optimization (see the last paragraph of the Discussion). In particular, our analysis suggests that crossing bottlenecks in the loss landscape, as opposed to convergence to critical points, may play a fundamental role in characterizing the final behavior of the network. This aspect has, until now, been largely ignored and we are hopeful it may be fruitfully integrated in the current understanding of deep networks using, for example, tools from non-equilibrium dynamics, where such studies are common.
>
> Although we agree on the need for an analytical model, we tried to avoid the pitfalls of prematurely settling on a particular abstraction of the problem in order to paint a clearer picture, both through empirical experiments and by establishing connections with the most recent theories in deep learning, and yet providing a novel approach where the Fisher Information becomes one of the central quantities to consider.
>
> >> To learn more a simpler toy model may be worthwhile to study.
>
> We fully agree. In this paper, we focused on testing our hypotheses on current state-of-the-art models and relatively complex datasets, in order to understand what are the key aspects that need to be captured by any simplified model. Now that this is established, and shown to be of practical relevance, given the widespread practice of fine-tuning, we can and will focus on simpler models that perhaps are also tractable analytically.

---

### Official Review · AnonReviewer2 · 2018-11-06
**Very good paper identifying a novel phenomenology in training of deep neural networks: the presence of "critical periods" (reminiscent of the same phenomenon in many biological brain circuits) where perturbations in training can permanently affect the final performance of the model.**

**Rating:** 8
**Confidence:** 4

**Review:**

The authors analyze the learning dynamics in deep neural networks and identify an intriguing phenomenon that reflects what in biological learning is known as critical period: a relatively short time window early in post-natal development where organisms become particularly sensitive to particular changes in experience. The importance of critical periods in biology is due to the fact that specific types of perturbations to the input statistic can cause deficits in performance which can be permanent in the sense that later training cannot rescue them.

The authors did a great job illustrating the parallelism between critical periods in biological neural systems and the analogous phenomenon in artificial deep neural networks. Essentially, they showed that blurring the input samples of the cifar10 dataset during the initial phase of training had an effect that is very reminiscent of the result of sensory deprivation during the critical periods of visual learning in mammals, resulting in a long-term impairments in visual object recognition that persists even if blurring is removed later in training. The authors go as far as characterizing the effects of the length of the "sensory deprivation" window and its onset during training, and comparing the results to classic neuroscience monocular deprivation experiments in kittens, pointing out very striking phenomenological similarities.

Next, the authors establish a connection between critical periods in deep neural networks and the amount of information that the weights of the trained model contain about the task by looking at the Fisher Information Matrix (FIM). With this method they obtain a host of interesting insights. One insight is that there are two phases in learning: an initial one where the trace of the FIM grows together with a rapid increase in classification accuracy, and a second one where accuracy keeps slightly increasing, but Fisher Information trace globally decreases. They then go into detail and look at how this quantity evolves within individual layers of the deep learning architecture, revealing that the deficit caused by the blurring perturbation during the early epochs training is accompanied by larger FIM trace in the last layers of the architecture at the expense of the intermediate layers.
Besides the fact that deep neural network exhibit critical periods, another important result of this work is the demonstration that pretraining, if done inappropriately can actually be deleterious to the performance of the network.

This paper is insightful, and interesting. The conceptual and experimental part of the paper is very clearly presented, and the methodology is very appropriate to tease apart some of the mechanisms underlying the basic phenomenological observations. Here are some detailed questions meant to elucidate some points that are still unclear.

- Presumably, early training on blurred images prevents the initial conv filters from learning to discriminate high-frequency components (first of all, is this true?). The crucial phenomenon pointed out by the authors is that, even after removing the blur, the lower convolutions aren't able to recover and learn the high-frequency components. In fact, the high FIM trace in the latest layers could be due to the fact that they're trying to compensate for the lack of appropriate low-level feature extractors by composing low-frequency filters so as "build" high-frequency ones. If this makes sense, one would assume that freezing the last layers and only maintaining plasticity in the lower ones could be a way of "reopening" the critical period. Is that indeed the case?
- The authors show that their main results are robust to changes in the learning rate annealing schedule. However, it is not clear how changing the optimizer might affect the presence of the critical period. What would happen for instance using Adam or another optimization procedure that relies on the normalization of the gradient?
- On a related note, the authors point out the importance of forgetting, in particular as the main mechanism behind the second learning phase. They also point out that the deficit in learning the task after sensory deprivation is accompanied by large FIM trace in the last layers. What would happen in the presence of a standard regularizer like weight decay? Assuming that large FIM trace in the last layers is correlated with large weighs, that might mitigate the negative effect of early sensory deprivation.
- In neuroscience the opening of the critical period window if thought to be mechanistically mediated by the maturation of inhibition. Is that view compatible with the results presented in this paper? This is sort of complementary to the FIM analysis, since is mostly about net average input to a neuron, i.e. about the information contained in the activations, rather than the weights.

---

> ### Author Response · Authors · 2018-11-21
> **Response to Reviewer 2**
>
> We are thankful to the reviewer for the feedback and the many insightful suggestions. We have added the suggested experiments to the revised version of the paper (in particular Figure 3, Figure 8), and also discuss some of the points more in detail below.
>
> >> Presumably, early training on blurred images prevents the initial conv filters from learning to discriminate high-frequency components (first of all, is this true?).  The crucial phenomenon pointed out by the authors is that, even after removing the blur, the lower convolutions aren't able to recover and learn the high-frequency components.
>
> We share the same intuition: We added to Figure 8 in the Appendix a visualization of the first-layer filters for networks with and without a deficit, and with the deficit removed after the end of the critical period. Figure 8 qualitatively shows that if high-resolution stimuli are not available before the critical period, the network does not manage to extract high-frequency features in the first layer. Unfortunately, the filters of the architecture we use are small (3x3), making the analysis more difficult: We are considering alternate experiments to test this hypothesis indirectly using responses to sinusoidal gratings to obtain clearer results.
>
> >> In fact, the high FIM trace in the latest layers could be due to the fact that they're trying to compensate for the lack of appropriate low-level feature extractors by composing low-frequency filters so as "build" high-frequency ones. If this makes sense, one would assume that freezing the last layers and only maintaining plasticity in the lower ones could be a way of "reopening" the critical period.  Is that indeed the case?
>
> This is a very interesting hypothesis: We tried to test it as suggested, freezing layers 3-6 of AllCNN while leaving layers 0-3 and the final classifier free to change. We observe that upon deficit removal the network error increases (since the data distribution changes), only to revert to the performance with deficit soon after. This may be because freezing the upper layers makes new information extracted by the lower layers invisible to the classifier, and therefore does not promote learning of new features. Rather, lower layers may adapt to blur them as before, to fit the upper response expected by the upper layers. However, we agree that finding ways to reopen critical periods (either by augmenting the data with more "stimulating" experiences, as is sometimes done in neuroscience, see e.g. Knudsen, J. Cogn. Neurosci., 2004, or by changing the training procedure) is an intriguing question.
>
> >> The authors show that their main results are robust to changes in the learning rate annealing schedule. However, it is not clear how changing the optimizer might affect the presence of the critical period. What would happen for instance using Adam or another optimization procedure that relies on the normalization of the gradient?
>
> We have conducted the experiment suggested and show in Figure 3 (Bottom Right) the result of training a ResNet with Adam, following the same experimental setup as Figure 1, and confirming that Adam also follows a similar trend.
>
> >> On a related note, the authors point out the importance of forgetting, in particular as the main mechanism behind the second learning phase. They also point out that the deficit in learning the task after sensory deprivation is accompanied by large FIM trace in the last layers. What would happen in the presence of a standard regularizer like weight decay?
>
> We fully agree on the importance of weight decay for critical periods, and in the revised version of the paper we have added new experiments that corroborate it (Figure 3, bottom left). We observe that training in the same setup as Figure 1, but without weight decay, leads to a sharper and sensibly shorter critical period.  Gradually increasing the value of weight decay leads to more prolonged critical periods, up to the point where the network eventually stops training properly altogether.

---

> > ### Author Response · Authors · 2018-11-21
> > **Response to Reviewer 2 (continued)**
> >
> > >> In neuroscience the opening of the critical period window if thought to be mechanistically mediated by the maturation of inhibition. Is that view compatible with the results presented in this paper? This is sort of complementary to the FIM analysis, since is mostly about net average input to a neuron, i.e. about the information contained in the activations, rather than the weights.
> >
> > We thank the reviewer for the insightful comment. Opening and closing of critical periods in neuronal networks have indeed been shown to be regulated by inhibitory (mostly GABAergic, but not exclusively) neuronal populations, which provide the critical balance between competing pathways (such as in ocular dominance) shaping the network in its mature form (Hensch, Curr. Top. Dev. Biol., 2005). While the CNN architectures we have tested in our study do not have direct inhibitory connections between elements of the CNN, we can speculate that "diffuse" inhibitory effects could emerge naturally during network optimization in order to make the inference more robust, leading to the effective "pruning" of certain connections, as mirrored by the FIM trace (Figure 4). It should also be noted here the connection existing between the decrease of the information in the weights (e.g., by "pruning") and the loss of information in the corresponding activations.
> >
> > In addition we have the fact that only datasets which provide robust "stimulation" of the CNNs exhibit critical-period-like behavior, while being fed noise as the input is a deficit that the network promptly recovers from (Figure 2, left). An analogous phenomenon has been demonstrated in kittens, with dark rearing causing prolonged plasticity and delayed critical period inception (and closure), but, remarkably, this behavioral evidence has been tied to the decreased inhibitory GABAergic tone in the relevant circuits (e.g. Chen et al., Mol. Brain Res. 2001), which, by not providing the necessary competitive balance, lengthens the plastic window of visual development.

---

### Official Review · AnonReviewer4 · 2018-11-08
**Interesting experiments casting some light on surprising properties of artificial neural networks**

**Rating:** 9
**Confidence:** 4

**Review:**

Let's be frank: I have never been a fan of comparing real brains with back-prop trained multilayer neural networks that have little to do with real neurons.  For instance, I am unmoved when Figure 1 compares multilayer network simulations with experimental data on actual kitten. More precisely, I see such comparisons as cheap shots.

However, after forgetting about the kitten,  I can see lots of good things in this paper.  The artificial neural network experiments designed by the authors show interesting phenomena in a manner that is amenable to replication. The experiments about the varied effects of different kinds of deficits are particularly interesting and could inspire other researchers in creating mathematical models for these striking differences.  The authors also correlate these effects with the two phases they observe in the variations of the trace of the Fisher information matrix.  This is reminiscent of Tishby's bottleneck view on neural networks, but different in interesting ways. To start with, the trace of the Fisher information matrix is much easier to estimate than Tishby's mutual information between patterns, labels, and layer activation. It also might represent something of a different nature, in ways that I do not understand at this point.

In addition the paper is very well written, the comments are well though, and the experiments seem easy to replicate.

Given all these qualities, I'll gladly take the kitten as well..

---

> ### Author Response · Authors · 2018-11-23
> **Response to Reviewer 4**
>
> We are thankful to the reviewer for their positive assessment of our paper. In fact, we share the same sentiment, as we articulate in the Conclusion, that one should resist the temptation to build too much on structural correspondences between such diverse systems. By showing these data we mostly wanted to emphasize how our reasoning is inspired by a reflection on the neurobiology of visual systems, and how such paradigms could be employed to better understand DNNs, since both systems share similar information processing goals.

---

### Public Comment · (anonymous) · 2018-11-05
**Questions About Novelty**

I wonder if there is any novelty in your experiment about Fisher Information. Many of the phenomenon in your experiments have been studied in https://arxiv.org/abs/1703.00810

---

> ### Author Response · Authors · 2018-11-05
> **The observed phenomena and their connection to irreversible phases of learning have not been studied and cannot be derived from the referenced paper**
>
> The phenomena observed in our study of the Fisher Information of the weights, and especially their connections with irreversible changes in the connectivity of Deep Networks during the early phases of training, cannot be derived from the results of Shwartz-Ziv and Tishby concerning the Shannon mutual information of the activations. In fact, to the best of our knowledge, we are the first to show the relationship between changes of Fisher Information and irreversible effects of optimizing a Deep Network. It is however true that some of the results by Shwartz-Ziv and Tishby are related, in fact implied, by our observations, which therefore provide further and independent corroborations of their claims.
>
> In particular, Shwartz-Ziv & Tishby report changes in the Shannon mutual information of the activations (not of the weights) during training. This are however not observed to be associated with any kind of irreversible changes. In fact, the existence of critical (irreversible) phases of learning have not been observed, let alone studied, by https://arxiv.org/abs/1703.00810 nor by anyone else to our knowledge. Furthermore, note that changes in mutual information of the activations are always observed during training, however critical learning periods are present only for some specific types of deficits. Therefore, Shwartz-Ziv & Tishby's framework and results on the different phases of information in the activations during training cannot explain the existence and the observed phenomenology of critical periods.
>
> On the other hand, thanks to the introducing of the Fisher Information of the weights, and by exploiting its relationship to the network connectivity, we can empirically characterize critical-period-inducing deficits as precisely those those that severely alter the connectivity of the network, and suggest a theoretical explanation for these phenomena (see end of Section 3). To the best of our knowledge, we are the first to compute and track these changes of the Fisher information of the weights during training of a state of the art, modern deep network, and in particular, nobody has shown plots like those in Figure 1, Figure 2, Figure 3, Figure 5.
>
> However, even if discussing different quantities, Shwartz-Ziv and Tishby’s results are indeed related to the plots we show in Figure 4, as we also discuss in Section 4 (pag. 7, second paragraph). The non-trivial connection between the two can be derived from the bound on information introduced by Achille and Soatto (https://arxiv.org/abs/1706.01350, JMLR 2018): As we describe on Page 7, they show that reduction of the information in the weights implies information reduction in the activation (but not vice-versa). In this sense, our results are can also serve to corroborate and expand, using an independent framework, the experimental evidence on the existence of multiple phases of learning shown by Shwartz-Ziv and Tishby.

---

### Meta-Review · Area_Chair1 · 2018-12-15
**Interesting observations about critical learning periods in deep networks in a well-written paper**

**Confidence:** 5
**Recommendation:** Accept (Poster)

**Metareview:**

Irrespective of their taste for comparisons of neural networks to biological organisms, all reviewers agree that the empirical observations in this paper are quite interesting and well presented. While some reviewers note that the paper is not making theoretical contributions, the empirical results in themselves are intriguing enough to be of interest to ICLR audiences.